# The Use of Calcaneal Quantitative Ultrasound as a Bone Health Screening Tool Amongst People Living with HIV and Taking Tenofovir-Based Antiretroviral Therapy: A Pilot Study

**DOI:** 10.3390/biomedicines13122847

**Published:** 2025-11-21

**Authors:** Wenjian Zhao, Muhamad Riduan Daud, Hashwin Singh Tejpal Singh, Sophia Ogechi Ekeuku, Xiong Khee Cheong, Najma Kori, Petrick Periyasamy, Kok-Yong Chin

**Affiliations:** 1Department of Pharmacology, Faculty of Medicine, Universiti Kebangsaan Malaysia, Bandar Tun Razak 56000, Malaysia; p112200@siswa.ukm.edu.my (W.Z.); hashwinsingh2000@gmail.com (H.S.T.S.); virgosapphire2088@yahoo.com (S.O.E.); 2Infectious Disease Unit, Department of Internal Medicine, Faculty of Medicine, Universiti Kebangsaan Malaysia, Bandar Tun Razak 56000, Malaysia; eewey_2104@yahoo.com (M.R.D.); cheongxk@gmail.com (X.K.C.); najmakori@ukm.edu.my (N.K.); petrickperiyasamy@gmail.com (P.P.)

**Keywords:** bone mineral density, osteopenia, osteoporosis, receiver-operating curve analysis, TDF

## Abstract

**Background/Objectives:** Individuals living with human immunodeficiency virus (HIV) receiving tenofovir-based antiretroviral therapy (ART) are at increased risk of bone loss. Dual-energy X-ray absorptiometry (DXA), the gold standard for determining bone mineral density (BMD), has limited accessibility. Calcaneal quantitative ultrasound (QUS) is an alternative screening tool for bone health, but its performance needs validation. The study aims to compare the performance of QUS between men with HIV on tenofovir-based ART and uninfected men. **Methods:** A cross-sectional study involving 52 men with HIV on tenofovir-based ART and 42 uninfected men was performed. Their bone health status was determined using DXA (lumbar spine and femoral neck) and QUS (calcaneus). The performance of QUS in identifying individuals with low BMD (DXA T-score < −1) was evaluated using receiver-operating characteristic curve analysis. **Results:** The HIV-ART group showed significantly lower QUS indices compared to the non-HIV-ART group (*p* < 0.05). The performance of QUS in identifying individuals with low BMD was poor in the HIV-ART group (*p* > 0.05), but moderate in the non-HIV-ART group (*p* < 0.05). After optimising the cutoffs, the performance of QUS was improved in the non-HIV-ART group but not in the HIV-ART group. **Conclusions:** QUS was not effective in identifying individuals with low BMD in men with HIV on ART. Its utility as a standalone screening tool in this population may be limited. Additional or more sensitive methods should be prioritised for bone health screening.

## 1. Introduction

Antiretroviral therapy (ART) has transformed human immunodeficiency virus (HIV) infection from a fatal disease into a manageable chronic condition [1]. However, the improved longevity of people living with HIV (PLWH) has led to an increased risk of comorbidities, including degenerative bone diseases. A significantly higher risk of low bone mineral density (BMD) and fractures is reported among PLWH compared to uninfected individuals [2,3]. Tenofovir-containing ART could contribute to bone loss among PLWH by directly altering bone cell energy metabolism and indirectly through tubulopathy and vitamin D metabolism [4,5,6,7]. A study in Japan reported fractures in young male PLHW after more than 5 years of treatment [8]. This observation highlights osteoporosis as an emerging healthcare concern in contemporary HIV care.

Dual-energy X-ray absorptiometry (DXA) is the gold standard for diagnosing osteoporosis [9], but its accessibility in resource-limited settings, where most PLWH reside, presents significant challenges [10]. DXA requires specialised equipment, trained technologists and dedicated facilities with radiation safety measures [11], representing substantial capital investment. The absence of DXA often results in inadequate bone health screening, delayed diagnosis of osteoporosis, and missed opportunities for fracture prevention in HIV populations [12]. The screening gap is particularly problematic given the younger age at which bone disease manifests in PLWH compared to the general population, necessitating earlier monitoring [13].

Quantitative ultrasound (QUS) has emerged as a promising alternative technology for assessing bone health. QUS devices, typically applied to the calcaneus, measure bone properties through ultrasound wave propagation, providing parameters such as broadband ultrasound attenuation (BUA), speed of sound (SOS), and composite ultrasound indices [14]. QUS measurements have been reported to predict osteoporosis and fracture risk, and are a reliable screening tool for bone health [15]. QUS devices are portable, radiation-free, relatively inexpensive, and require minimal operator training [14], making them suitable for low-resource clinical settings. The absence of ionising radiation also makes QUS suitable for repeated assessments in younger patients and pregnant mothers [16,17].

Recent studies have demonstrated the utility of QUS in HIV populations, showing consistent associations between reduced calcaneal ultrasound parameters and HIV infection status. Cross-sectional analyses have revealed significantly lower QUS values in PLWH compared to HIV-negative controls, particularly in patients receiving specific ART regimens [18,19]. Importantly, several studies have documented moderate to strong correlations between calcaneal QUS measurements and DXA-derived BMD values, suggesting the potential of QUS as a screening tool [20,21].

Despite promising initial evidence, several critical gaps remain in the application of QUS in HIV care. QUS machine technology and reference values are manufacturer-specific and cannot be interpreted interchangeably between different machines. T-scores and Z-scores calculated by QUS are not the same as DXAs and should not be used for diagnosis directly [22]. The deployment of QUS in a local setting requires prior performance validation against DXA to establish diagnostic thresholds [23].

Given this research gap, the primary objective of this pilot study was to determine and compare the diagnostic performance of a calcaneal QUS device against DXA in two groups: (1) adult men living with HIV and receiving tenofovir-based ART and (2) uninfected adult men. The selection of younger adult men in this study reflects the demographics of PLWH in Malaysia, which are skewed towards men aged 20–49 years [24]. The inclusion of uninfected men helps us to assess whether a separate diagnostic cutoff is necessary. Understanding the utility and limitations of QUS in PLWH is essential for developing evidence-based screening strategies that can improve bone health outcomes while being practical and sustainable in diverse clinical settings.

## 2. Materials and Methods

### 2.1. Ethical Consideration

The protocol of this study was reviewed and approved by the Universiti Kebangsaan Malaysia Ethics Board (approval code: JEP-2024-665). Subjects provided written informed consent before participating in the study.

### 2.2. Subjects

This cross-sectional study used a purposive sampling technique to recruit subjects from June 2024 to July 2025. Patients living with HIV and taking tenofovir-based ART (the HIV-ART group) were recruited from the Infectious Disease Clinic, Hospital Canselor Tuanku Muhriz (HCTM), Kuala Lumpur, Malaysia. The included subjects were aged at least 21 years and had received ART for at least one year. Subjects previously diagnosed with bone metabolic diseases (osteoporosis, osteomalacia and rickets), rheumatoid arthritis, thyroid diseases, hypogonadism and chronic kidney disease greater than stage 3a were excluded. Subjects using pharmacological agents that modified bone metabolism, such as anti-osteoporosis drugs, glucocorticoids, anabolic steroids, anticonvulsants, and sex hormone replacement/deprivation therapy, were also excluded.

Healthy subjects were recruited amongst the staff of HCTM within the same period. They were neither exposed to HIV nor using ART (non-HIV-ART group). The other inclusion and exclusion criteria previously mentioned also applied to this group.

### 2.3. Data Collection

The age of the subjects was determined from their identification card records. Ethnicity was declared by the subjects. The height of the subjects without shoes was determined by a measuring tape and recorded to the nearest 1 cm. The weight of the subjects in light clothing was measured using a body composition analyser (ACCUNIC BC353, Accunic, Yuseong-gu, Republic of Korea) and recorded to the nearest 0.1 kg. The body mass index (BMI) was calculated according to the convention. The smoking and alcohol-drinking habits were self-declared.

The medical and medication history of the subjects was retrieved from the electronic medical record system (EMRS) of HCTM or self-declared. Since the HIV-ART group was followed up in HCTM, data on their HIV diagnosis and ART history were available in the EMRS.

### 2.4. Bone Health Assessment

The bone mineral density of the subjects was determined using the Hologic Discovery QDR Wi densitometer (Marlborough, MA, USA). The regions of interest were the femoral neck and the lumbar spine (L1-L4). T-scores and Z-scores were calculated based on the manufacturer’s reference for Asian populations. A single technician with no knowledge of the subjects’ HIV status completed all the scans.

The osteoporosis index (OI) of the subjects was determined using a calcaneal bone ultrasonometer (OSTEOKJ3000, Kejin, Nanjing, China). The coefficient of variation in the machine is 5% as indicated by the manufacturer. OI is the manufacturer-specific composite index for this device, conceptually similar to the ‘stiffness index’ (SI) reported for other QUS devices. The ultrasound signals that travelled through the calcaneal bones of the subjects were recorded by the device, and the speed of sound (SOS) and broadband ultrasound attenuation (BUA) were generated. OI, as a composite index, was calculated automatically using the following formula: OI = 0.106 × SOS + 0.5 × BUA − 127.4. OI values were converted to Z-scores and T-scores based on the manufacturer’s internal Asian reference. A different technician with no knowledge of the DXA results and HIV status of the subjects completed all the scans. DXA and QUS measurements were performed on the same day of the visit.

For bone health classification, the International Foundation of Osteoporosis suggested that a T-score < −1 should be used even for younger adults [25]. These cutoffs are not applicable to QUS, but they are used arbitrarily to indicate increased risk for low BMD.

### 2.5. Statistical Analysis

The normality of the continuous data was determined by the Shapiro–Wilk test. Continuous data were expressed as means and standard deviations. Categorical data were expressed as counts and percentages.

The comparison of basic demographic factors between patients living with HIV receiving ART (HIV-ART) and subjects not exposed to HIV and ART (non-HIV-ART) was performed using an independent *t*-test for continuous variables and a chi-square test for categorical variables. The continuous bone health parameters between the two groups were compared using univariate analysis, adjusting for age, ethnicity, BMI, and smoking and alcohol-drinking habits. Spearman’s correlation was used to determine the correlation between years of ART treatment and all bone health indices. Pearson’s correlation was used to determine the correlation between BMI and all bone health indices.

Kappa statistics were used to determine the diagnostic agreement between DXA and QUS. Per convention, a kappa value between 0.01 and 0.20 is slight, 0.21–0.40 is fair, 0.41–0.60 is moderate, 0.61–0.80 is substantial, and 0.81–1.00 is perfect agreement. Receiver-operating-curve analysis was used to determine the performance of QUS in identifying subjects with low BMD (T-score < −1), in terms of sensitivity, specificity, and area under the curve (AUC). Per convention, an AUC value between 0.50 and 0.70 is poor, 0.70–0.80 is moderate, 0.80–0.90 is good, and 0.90–1.00 is excellent discrimination. A decision for new cutoffs for QUS was made based on Youden’s index (J = sensitivity + specificity − 1).

All analyses were performed using SPSS version 26.0 (IBM, Armonk, NY, USA), with the statistical significance set at *p* < 0.05.

## 3. Results

The study initially recruited 56 HIV-ART and 56 non-HIV-ART subjects. However, only 52 HIV-ART and 42 non-HIV-ART subjects completed the screening procedures. All dropouts indicated constraints in time to complete both bone health assessments. The HIV-ART group had a median HIV diagnosis of 6 years (IQR 4 years) and a median ART treatment period of 5 years (IQR 4 years).

There was no significant difference in age, ethnicity, weight, height, BMI, and proportion of smokers and alcohol users (*p* > 0.05) between the two groups. For the DXA parameters, there was no significant difference in spine and femoral neck BMD and T-score after adjusting for age, ethnicity, BMI, and smoking and alcohol-drinking habits. For the QUS parameters, the BUA (*p* = 0.001) and QUS T-scores (*p* = 0.020) of the non-HIV-ART group were significantly higher compared to those of the HIV-ART group after adjusting for confounders. SOS and OI were not significantly different between the groups (*p* > 0.05). The bone health status (T-score < −1 or ≥−1) was also not significantly different between the two groups (*p* > 0.05) (Table 1). Correlation analysis revealed a significant positive association between BMI and DXA’s bone health indices, but no significant association between the duration of ART treatment and all bone health indices was found (Appendix A).

At a T-score of < −1, QUS showed poor performance in identifying subjects with low BMD (DXA T-score < −1), regardless of HIV-ART status (Figure 1). The AUC of the non-HIV-ART group (0.686, 95% CI: 0.525–0.858, *p* = 0.039) was greater than that of the HIV-ART group (0.545, 95% CI: 0.387–0.702, *p* = 0.387) and the overall subjects (0.609, 95% CI: 0.495–0.722, *p* = 0.071) (Table 2). The sensitivity, specificity, positive predictive value, and negative predictive value also followed similar trends (Non-HIV-ART > Overall > HIV-ART) (Table 3). The agreement between QUS and DXA at a T-score of < −1 was fair and significant in the non-HIV-ART group (κ = 0.357, *p* = 0.009) and the overall subjects (κ = 0.206, *p* = 0.013), but not in the HIV-ART group (κ = 0.085, *p* = 0.369) (Table 3).

Calculation of the optimum QUS T-score was performed. Based on Youden’s index, the best QUS T-score cutoff values to differentiate subjects with low BMD (DXA T-score < −1) were −1.15 for the non-HIV-ART group, −1.525 for the HIV-ART group, and −0.095 for the overall subjects. Based on the new cutoff values (Figure 2), the AUC for the non-HIV-ART group improved to 0.752 (95% CI: 0.601–0.902, *p* = 0.005). The AUC for HIV-ART (0.640, 95% CI: 0.488–0.792, *p* = 0.084) was similarly improved but did not reach statistical significance (Table 4). Specificity, PPV and NPV were improved in both groups, but not the sensitivity values (Table 4). No significant improvements in AUC and all diagnostic indices were observed for the overall subjects, as the new cutoff value was close to the original cutoff value (Table 4). The agreement between QUS and DXA at the new cutoff values improved for both the non-HIV-ART group (κ = 0.488, *p* = 0.001) and the HIV-ART group (κ = 0.276, *p* = 0.043). However, this improvement was not observed when the two groups were combined (Table 5).

## 4. Discussion

The current study found a significantly lower BUA and QUS T-score in the HIV-ART group compared to the non-HIV-ART group, despite the lack of a significant difference in their DXA bone parameters. QUS also demonstrated poor performance in identifying individuals with low BMD in the HIV-ART group. The performance of QUS in this regard could not be significantly improved by modifying the cutoff values.

The HIV-ART group has lower BUA and QUS T-scores than non-HIV-ART subjects, without corresponding changes in BMD parameters. This observation suggests that QUS and DXA reflect different bone properties, and ART-related changes can alter QUS indices more than DXA indices. QUS-derived parameters are associated with bone microarchitectural and mechanical properties [26,27], whereas DXA measures areal mineral content and does not capture microstructural deterioration or material properties. PLWH had been reported to experience deterioration in trabecular but not cortical microarchitectural parameters [28,29]. The subjects in this study might have experienced these changes, which were captured by QUS but not by DXA. Additionally, QUS measurements were performed at the calcaneus of the subjects, which has a higher trabecular content compared to the lumbar spine and femoral neck [30]. In metabolic bone disease, the trabecular-rich bone site can be disproportionately affected [31], resulting in lower QUS values. Since most demographic, anthropometric and lifestyle factors were similar between HIV-ART and non-HIV-ART groups, and have been adjusted in the analysis, they are less likely to confound the results.

The lack of a significant difference in BMD measured by DXA and prevalence of low bone health (T-score < −1) between the HIV-ART and non-HIV-ART groups was in contrast with previous findings [32,33,34]. Similarly, the duration of ART was not significantly correlated with bone health indices in the HIV-ART group, which disagreed with some previous reports [35]. This may be attributable to the relatively young age of our cohort (mean 43 years) and the moderate duration of ART (median 5 years). Some reports suggest that adjusting body weight and BMI, as in this study, would reduce the difference [32,33]. It is also possible that DXA, which measures areal BMD, is less sensitive than QUS to detect early bone changes in this demographic. A retrospective cohort study among Asian ART users with chronic hepatitis indicated no significant difference in the incidence of osteoporosis across 8 years, indicating a limited impact of ART on bone health [36].

The performance of QUS in identifying individuals with low BMD was moderate in the non-HIV-ART group but improved with the new cutoff. This observation was comparable to a previous Malaysian study on middle-aged and elderly individuals, with the AUC, sensitivity, and specificity of the male subjects reported as 0.657 (95% CI: 0.602–0.712), 75.6%, and 55.8% [23]. However, the poor performance of QUS in this regard was unexpected, as some previous studies had reported high correlations between QUS and DXA measurements [21]. The performance of QUS remained poor, even with the cutoff modification in this group. Other studies did not report agreement with DXA and did not perform performance analysis, similar to the current study [18,19,20,37]. The reasons for the differences in QUS performance between the HIV-ART and non-HIV-ART groups were not investigated in the study. We postulated that soft tissue differences exist between HIV-ART and non-HIV-ART individuals, which affect the agreement between QUS and DXA. Oedema and fat tissue thickness are factors known to reduce QUS values [38,39,40]. ART has been reported to cause lipodystrophy in PLWH, and this alteration could affect QUS readings [41]. However, we did not perform an assessment of ankle oedema and fat distribution, so this reason remains speculative.

The poor performance of QUS in identifying individuals with low BMD in the HIV-ART group, as observed in this study, does not nullify its role in bone health screening for this population. The readers should note that this is a small-scale pilot study, and the findings should be validated in a more comprehensive study, taking into consideration confounders such as ankle oedema and fat distribution. Readers should also be aware that QUS measurements are manufacturer-specific, and the results may not be directly translatable to results from another study. The subjects of this study might not be representative of the HIV-ART population in Malaysia due to the single-centred design and non-randomised sampling method used. The cross-sectional design also prevents us from investigating the causal effects of ART on bone deterioration in PLWH. Other important confounders of bone health, such as sedentary lifestyle, vitamin D levels, and dietary records of the subjects, were not collected in the study. Thus, their effects cannot be adjusted for in the analysis. Nevertheless, this is the first study that investigates the performance of QUS as a bone health screening tool for PLWH on ART in Malaysia, which may lead to a more comprehensive study in the future.

Since this is a pilot study, an a priori sample size calculation was not performed. However, using the results derived from the current study, with an expected AUC of 0.7, prevalence of low bone health based on lumbar spine (DXA T-score < −1) of 0.157, a type 1 error rate of 0.05, and a power of 0.8, we calculated that 300 subjects will be required for each group in the future study [42]. This number exceeds the number of PLWH followed at the authors’ institution; thus, a multicenter recruitment strategy will be required.

## 5. Conclusions

This study found that the performance of QUS was not satisfactory as an independent screening tool for low BMD in men with HIV receiving ART, even though this population has lower QUS bone health indices than healthy controls. Misclassification of bone health status could occur if judgement was based solely on QUS. PLWH with strong risk factors of bone loss should be referred for a DXA scan for early detection. The role of QUS needs to be validated in a more comprehensive study in the future.

## Figures and Tables

**Figure 1 biomedicines-13-02847-f001:**
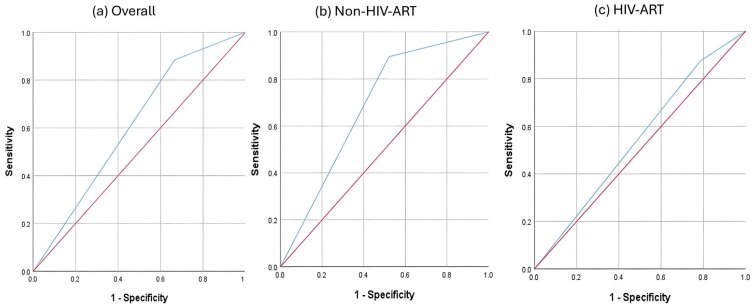
ROC analysis of QUS T-score (<−1) in identifying individuals with low BMD (DXA T-score < −1). Notes: Red line—reference line representing 50% area under the curve. Blue line—performance of calcaneal quantitative ultrasound.

**Figure 2 biomedicines-13-02847-f002:**
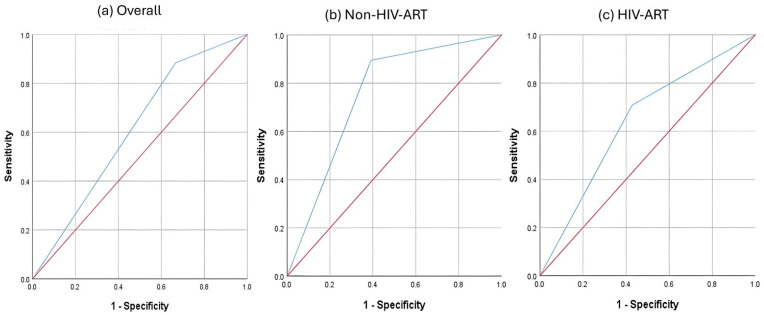
ROC analysis of QUS T-score (new cutoff values) in identifying individuals with low BMD (DXA T-score < −1). Notes: Red line—reference line representing 50% area under the curve. Blue line—performance of calcaneal quantitative ultrasound.

**Table 1 biomedicines-13-02847-t001:** Characteristics of the subjects.

Characteristics		Non-HIV-ART		HIV-ART		*p*-Value
		Mean	SD	Mean	SD	
	Age (years)	43.0	8.9	43.0	8.0	0.991
	Weight (kg)	73.4	10.8	73.5	13.0	0.952
	Height (cm)	170.1	5.9	170.7	5.3	0.590
	BMI (kg/m^2^)	25.4	3.4	25.3	4.4	0.890
DXA indices	Lumbar spine BMD (g/cm^2^) *	0.99	0.14	0.97	0.13	0.376
	Femoral neck BMD (g/cm^2^) *	0.77	0.11	0.76	0.11	0.418
	Lumbar spine T-score *	0.20	1.06	0.02	1.03	0.374
	Femoral neck T-score *	−0.57	1.16	−0.83	1.23	0.237
US indices	SOS (m/s) *	1525.66	23.14	1521.73	20.98	0.349
	BUA (dB) *	28.02	7.09	23.79	5.39	0.002
	OI *	46.70	5.83	45.45	4.46	0.205
	T-score *	−1.27	0.92	−1.70	0.89	0.022
		**n**	**%**	**n**	**%**	** *p* ** **-Value**
Race	Chinese	16	38.1	18	34.6	0.940
	Indian	2	4.8	3	5.8	
	Malay	24	57.1	31	59.6	
Cigeratte smoking	Smokers	34	81	38	73.1	0.259
	Non-smokers	8	19	14	26.9	
Alcohol drinking	Regular alcohol drinkers	34	81	44	84.6	0.421
	Non-regular alcohol drinkers	8	19	8	15.4	
Viral load	Detectable	n/a	n/a	10	19.2	n/a
	Not detectable	n/a	n/a	42	80.8	
DXA lumbar spine T-score	≥−1	35	83.3	44	84.6	0.866
	<−1	7	16.7	8	15.4	
DXA femoral neck	≥−1	24	57.1	29	55.8	0.894
	<−1	18	42.9	23	44.2	
QUS	≥−1	13	31	9	17.3	0.120
	<−1	29	69	43	82.7	

Abbreviations: BMD, bone mineral density; BMI, body mass index; BUA, broadband attenuation of sound; n/a, not available; QUS, quantitative ultrasound; OI, osteoporosis index; SD, standard deviation; SOS, speed of sound. Notes: * denotes a comparison adjusted for age, ethnicity, BMI, smoking and alcohol drinking habits.

**Table 2 biomedicines-13-02847-t002:** Performance of QUS T-score (<−1) in identifying individuals with low BMD (DXA T-score < −1).

Group	AUC	SE	*p*	Asymptotic 95% CI		Sensitivity	Specificity	PPV	NPV
				Lower Bound	Upper Bound				
Non-HIV-ART	0.686	0.083	0.039	0.525	0.848	0.895	0.478	0.586	0.846
HIV-ART	0.545	0.08	0.582	0.387	0.702	0.875	0.214	0.488	0.667
Overall	0.609	0.058	0.071	0.495	0.722	0.884	0.333	0.528	0.773

Abbreviations: AUC, area under the curve; CI: Confidence Interval; HIV-ART, men with HIV receiving ART; non-HIV-ART, uninfected men; NPV, negative predictive value; PPV, positive predictive value; SE, standard error.

**Table 3 biomedicines-13-02847-t003:** Agreement in low BMD classification by QUS and DXA.

Group	Technique	Bone Health Status	DXA				
			Normal	Low BMD	Total	Kappa	*p*-Value
Non-HIV-ART	QUS	Normal	11	2	13	0.357	0.009
		Low BMD	12	17	29		
	Total		23	19	42		
HIV-ART	QUS	Normal	6	3	9	0.085	0.369
		Low BMD	22	21	43		
	Total		28	24	52		
Overall	QUS	Normal	17	5	22	0.206	0.013
		Low BMD	34	38	72		
	Total		51	43	94		

Abbreviations: BMD, bone mineral density; HIV-ART, men with HIV receiving ART; non-HIV-ART, uninfected men; QUS, quantitative ultrasound.

**Table 4 biomedicines-13-02847-t004:** Performance of QUS T-score (new cutoff values) in identifying individuals with low BMD (DXA T-score < −1).

Group	Cutoff	J-Index	AUC	SE	*p*	Asymptotic 95% CI		Sensitivity	Specificity	PPV	NPV
						Lower Bound	Upper Bound				
Non-HIV-ART	−1.15	0.504	0.752	0.077	0.005	0.601	0.902	0.895	0.609	0.654	0.875
HIV-ART	−1.525	0.279	0.64	0.078	0.084	0.488	0.792	0.708	0.571	0.586	0.696
Overall	−0.095	0.217	0.609	0.058	0.071	0.495	0.722	0.884	0.333	0.528	0.773

Abbreviations: AUC, area under the curve; CI: Confidence Interval; HIV-ART, men with HIV receiving ART; J-index, Youden’s index; non-HIV-ART, uninfected men; NPV, negative predictive value; PPV, positive predictive value; SE, standard error.

**Table 5 biomedicines-13-02847-t005:** Agreement in low BMD classification by QUS (new cutoff values) and DXA (T-score < −1).

Group	Technique	Bone Health Status	DXA				
			Normal	Low BMD	Total	Kappa	*p*-Value
Non-HIV-ART	QUS	Normal	14	2	16	0.488	0.001
		Low BMD	9	17	26		
	Total		23	19	42		
HIV-ART	QUS	Normal	16	7	23	0.276	0.043
		Low BMD	12	17	29		
	Total		28	24	52		
Overall	QUS	Normal	17	5	22	0.206	0.013
		Low BMD	34	38	72		
	Total		51	43	94		

Abbreviations: BMD, bone mineral density; HIV-ART, men with HIV receiving ART; non-HIV-ART, uninfected men; QUS, quantitative ultrasound.

## Data Availability

Data are available upon reasonable request from the corresponding author. The data are not publicly available due to the sensitivity of the subjects investigated (PLWH).

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
