# Peer review of "The Use of Calcaneal Quantitative Ultrasound as a Bone Health Screening Tool Amongst People Living with HIV and Taking Tenofovir-Based Antiretroviral Therapy: A Pilot Study"

_biomedicines, 2025, doi:10.3390/biomedicines13122847_

Round 1

Reviewer 1 Report

Comments and Suggestions for Authors

Thank you for asking me to review this paper from Dr Wenjian Zhao et al from Malaysia. This study is interesting, few items should be addressed:

  • Introduction is too long. It should be shortened.
  • The objective of this study should be clearly explained.
  • According to guidelines, obesity, sedantary life style and vitamin D deficiency is associated with increased risk of osteoporosis. The authors should mention obesity and vitamin D deficiency in method section.
  • There is no information regarding life style intervention.
  • The table 4 and 5 are entirely meaningless.
  • Some orthographic and grammatical errors need to be corrected throughout the manuscript.

The following papers should be cited an discussed:

  • Casado JL, Santiuste C, Vazquez M, et al (2016) Bone mineral density decline according to renal tubular dysfunction and phosphaturia in tenofovir-exposed HIV-infected patients. AIDS (London, England) 30:1423-31.
  • Komatsu A, Ikeda A, Kikuchi A, Minami C, Tan M, Matsushita S (2018) Osteoporosis-Related Fractures in HIV-Infected Patients Receiving Long-Term Tenofovir Disoproxil Fumarate: An Observational Cohort Study. Drug safety 41:843-8.
  • Wei MT, Le AK, Chang MS, et al (2019) Antiviral therapy and the development of osteopenia/osteoporosis among Asians with chronic hepatitis B. Journal of medical virology 91:1288-94.
  • Côté HC (2007) Mechanisms of antiretroviral therapy-induced mitochondrial dysfunction. Current opinion in HIV and AIDS 2:253-60.
  • Grigsby IF, Pham L, Mansky LM, Gopalakrishnan R, Mansky KC (2010) Tenofovir-associated bone density loss. Therapeutics and clinical risk management 6:41-7.
  • Grant PM, Cotter AG (2016) Tenofovir and bone health. Current opinion in HIV and AIDS 11:326-32.
  • Kohler JJ, Hosseini SH, Hoying-Brandt A, et al (2009) Tenofovir renal toxicity targets mitochondria of renal proximal tubules. Laboratory investigation; a journal of technical methods and pathology 89:513-9.

Comments on the Quality of English Language

Some orthographic and grammatical errors need to be corrected throughout the manuscript.

Author Response

Thank you for reviewing our manuscript. We have carefully considered your comments and have responded to each of them in the attached response sheet.

Reviewer 2 Report

Comments and Suggestions for Authors

I have a few observations.

  1. Because of the cross-sectional study design, sampling method, data presented, number of confounders unaddressed in the analysis such as viral load, physical activity, vitamin D levels, calcium supplementation and other dietary habits, I suggest editing the last line of the abstract. The line '' It should not be used alone for bone health screening in this population'' reads a bit too strong, too definite. This line can be rewritten to suggest using additional or more sensitive tool in this population.
  2. Do we have data on viral loads of the HIV-ART cases?
  3. In determining sample size, what was the power and what the targeted effect size? This may be helpful in interpreting these results.
  4. What was the interval between the QUS and DEXA measures?
  5. Some studies have shown that the Stiffness Index (SI) performs better than the BUA and SOS individually. Is the Osteoporosis Index here the same as the Stiffness Index?

Author Response

(The authors gave the same response as above.)

Reviewer 3 Report

Comments and Suggestions for Authors

The  manuscript.” The use of calcaneal quantitative ultrasound as a bone health  screening tool amongst people living with HIV and taking  tenofovir-based antiretroviral therapy: A pilot study ”  by Zhao W  et al. aimed to compare the performance of QUS between men with HIV on tenofovir based ART and uninfected men.   The conclusion was that QUS was not effective in identifying individuals with low BMD in men with HIV on ART.

COMMENTS

1).  Authors should indicate the reasons why 14 non-HIV individuals did not complete the screening procedures.

2). In this study, BMD values measured by DXA in HIV-infected patients who had been on long-term therapy were essentially identical to those of healthy controls. This finding contrasts with most reports in the literature and therefore warrants further discussion.

3).  For a better understanding of the results, the authors should report the number of HIV ART and non-HIV ART subjects with T-score values ​​>-1 SD as being normal, between -1 SD and -2.5 SD as indicating osteopenia, and ≤-2.5 SD as indicating osteoporosis for both QUS and DXA techniques.

4). It would be interesting to evaluate the correlation between the T-score values ​​by QUS and BY DXA and the BMI and the years of anti-HIV therapy.

5). The overall quality of tables and figures needs to be improved.

6). The precision characteristics of OSTEOKJ300 should be reported.

Author Response

(The authors gave the same response as above.)

Round 2

Reviewer 1 Report

Comments and Suggestions for Authors

        The study design is straightforward.

  • English language fine. No issues detected.
  • Overall, I would recommend publication of the manuscript because the amendments were made.

Comments on the Quality of English Language

  • English language fine. No issues detected.

Reviewer 3 Report

Comments and Suggestions for Authors

I read the revised manuscript. The Authors adequately responded to the comments of the reviewer.

My decision is : Accept.